# Joint Fine-tuning and Conversion of Pretrained Speech and Language Models towards Linear Complexity

**Mutian He**[1,2]**, Philip N. Garner**[1]
[1] Idiap Research Institute, Martigny, Switzerland
[2] Ecole Polytechnique Fédérale de Lausanne (EPFL), Switzerland
{mutian.he,phil.garner}@idiap.ch

## Abstract

Architectures such as Linformer and Mamba have recently emerged as competitive linear time replacements for transformers. However, corresponding large pretrained models are often unavailable, especially in non-text domains. To remedy this, we present a Cross-Architecture Layerwise Distillation (CALD) approach that jointly converts a transformer model to a linear time substitute and fine-tunes it to a target task. We also compare several means to guide the fine-tuning to optimally retain the desired inference capability from the original model. The methods differ in their use of the target model and the trajectory of the parameters. In a series of empirical studies on language processing, language modeling, and speech processing, we show that CALD can effectively recover the result of the original model, and that the guiding strategy contributes to the result. Some reasons for the variation are suggested.

## 1 Introduction

Recent progress in language modeling from BERT (Devlin et al., 2019) to Llama3 (Dubey et al., 2024) has witnessed the rise of the attention mechanism and transformer architecture. Such models are able to scale up to very large model capacity and pretraining data for natural language processing (NLP), and have lead to breakthroughs in a broad range of downstream applications under either fine-tuning or zero-shot/in-context scenarios. This is surely not restricted to the text domain: Off-the-shelf speech and audio models like Wav2Vec2 (Baevski et al., 2020), HuBERT (Hsu et al., 2021), and WavLM (Chen et al., 2022) are pretrained on extensive audio data; they can be fine-tuned to obtain good performance on various tasks from automatic speech recognition (ASR) to spoken language understanding (SLU). However, such large-scale models are computationally expensive; the main reason is that in the standard attention mechanism, each token interacts with all other tokens in the input sequence, forming a complete graph. The computation time therefore increases quadratically with the input length, and a linear-growing KV-cache is required. As a result, the computation requirements can become prohibitive, especially for long contexts. A large cohort of architectures and algorithms have been proposed to mitigate the problem and to reach linear complexity, using techniques such as low-rank projection, hashing, and sparse attention (Tay et al., 2023). More recently, there has been a revival of recurrent networks such as state space models (Gu et al., 2022; Gu & Dao, 2024), RWKV (Peng et al., 2023), and RetNet (Sun et al., 2023). By recurrently updating states, those models allow linear time and constant space inference while retaining the transformer's parallel training capability. It has been demonstrated that the latest recurrent models, notably Mamba2 (Dao & Gu, 2024), are capable of reaching performance comparative with the most well-trained transformer, even in large-scale models.

In spite of such transformer substitutes, there remain obstacles to putting them into practical use. These new models are not simply plug-and-play upgrades for existing pretrained models, but are generally pretrained from scratch again. The pretrained weights of these models are often not publicly released; the data and computational cost of repeating such pretraining for every new model can be prohibitive for users and researchers of downstream applications. In particular, most models are primarily developed for processing language or text. These models are typically general-purpose

substitutes for the standard attention mechanism, also applicable to other domains such as speech and audio. However, the pretrained models on those latter domains simply never exist; this hinders those domains from utilizing the latest models and algorithms created in the larger machine learning (ML) and NLP community. This is a particular concern for research in end-to-end speech processing involving long-distance dependency in the lower information density audio form: such models are a prerequisite to efficiently process the abundant long-form but poorly segmented audio in the wild.

To address this issue, a possible solution is to make use of off-the-shelf pretrained models and convert them into the target linear-complexity model. Several previous works have explored this method to obtain a target model with much less computation. Many of the standard transformer layers (e.g., embeddings and feed-forward) are still part of the target architecture. Compared to pretraining from scratch, an immediate improvement would be the reuse of those pretrained layers (Kasai et al., 2021; Choi, 2024). In addition to such parameter transfer, we can further guide the conversion by transferring model behavior using knowledge distillation (KD). Given model inputs, KD aligns the model outputs or hidden states between the original transformer (teacher) and the target model (student) using an extra loss term. In an earlier work, Tang et al. (2019) explored conversion from BERT to RNN by aligning outputs; more recently, Zhang et al. (2024a) and Bick et al. (2024) tried to reproduce the attention weights of the standard attention mechanism in linear attention models. However, it has been pointed out that reproducing attention weights is not a prerequisite to reproduce performance (Mao, 2022), and even causes training inefficiency and instability (Mercat et al., 2024). More importantly, this method is restricted to target models for which we can easily determine an attention weight matrix equivalent to that of the standard one. This is not always the case: e.g., the matrix in Linformer has a lower rank (Wang et al., 2020). Furthermore, distillation only on outputs may not be sufficient to guide the model. The above works are restricted to NLP tasks without considering other more demanding domains, e.g., speech. Furthermore, most of these works are focused on conversion using an efficient re-pretraining stage, which is still difficult for downstream users with limited access to computational resources and pretraining data. Instead, it would be more attractive to directly carry out the conversion in the fine-tuning phase.

Given the considerations above, we explore the possibility of converting an existing pretrained transformer into a linear-complexity model on the downstream task. Our approach broadly covers different source models, target models, and target tasks. This is made possible by combining parameter transfer and layerwise distillation with hidden states from a target teacher model, i.e., a pretrained standard transformer fine-tuned for the target task. In addition to direct guidance from hidden states of the target teacher, we further investigate two modes of guidance: Recognizing the importance of the pretrained knowledge of the original transformer that can be lost during fine-tuning, we can guide the converted model by the sequence of models on the teacher's fine-tuning trajectory. Alternatively, we can use the distilled model as a good initialization for conversion, and allow deviation from the teacher's behavior in the later training phase.

We demonstrate the advantage of our novel Cross-Architecture Layerwise Distillation (CALD) approach under a broad range of scenarios, including conversion from RoBERTa to Linformer for NLP tasks, from Pythia to Mamba for language modeling, and from Wav2Vec2 to Mamba2 for speech tasks including ASR, SLU, and speaker identification (SID). In all of these scenarios, our converted models demonstrate minor to no performance loss compared to the standard transformer, much better than naive parameter transfer. Code and resources to reproduce our work are available at `https://github.com/idiap/linearize-distill-pretrained-transformers`.

## 2    RELATED WORK

### 2.1    LINEAR-COMPLEXITY AND RECURRENT MODELS

**Linear-complexity models**. There is a large body of work on reducing the $O(N^2)$ time complexity of transformer owing to the attention computation $A(\boldsymbol{Q}, \boldsymbol{K}, \boldsymbol{V}) = \text{softmax}(\frac{\boldsymbol{Q}\boldsymbol{K}^T}{\sqrt{d}})\boldsymbol{V}$. The $N \times d$ matrices $\boldsymbol{Q}$, $\boldsymbol{K}$, and $\boldsymbol{V}$ represent the sequence of $N$ *query*, *key*, and *value* feature vectors, each with constant $d$ dimensions. A straightforward approach is to limit the attention to several pre-specified positions in the sequence, e.g., neighboring tokens, several special global tokens, or a sparse subset of the sequence (Child et al., 2019; Beltagy et al., 2020; Ainslie et al., 2020; Zaheer et al., 2020). Alternatively, the attended positions can be determined on-the-fly by hashing and clustering (Kitaev

et al., 2020; Roy et al., 2021). With Linformer, Wang et al. (2020) further found that the attention matrix $\boldsymbol{A}$ is inherently low-rank, and projects $\boldsymbol{K}$ and $\boldsymbol{V}$ into a lower $k$-dimensional space using $k \times N$ matrices $\boldsymbol{E}$ and $\boldsymbol{F}$. The two matrices can be shared, or even shared between all layers. Then, the $O(N^2)$ matrix multiplication can be avoided, as $\boldsymbol{A}$ can be provably approximated by

$$\tilde{A}(\boldsymbol{Q}, \boldsymbol{K}, \boldsymbol{V}) = \text{softmax}(\frac{\boldsymbol{Q}(\boldsymbol{EK})^T}{\sqrt{d}})\boldsymbol{FV}. \tag{1}$$

**Recurrent models**. By applying a kernel trick in a reverse direction, Katharopoulos et al. (2020) reduce transformers to RNNs of infinite dimension. In this RNN, hidden states $\boldsymbol{z}_t = \sum_{i \leq t} \phi(\boldsymbol{K}_{i,:})$ are the sum of an infinite-dimensional feature $\phi(\boldsymbol{K}_{i,:})$ mapped from each key vector (more specifically, along with their product with values). Using various finite dimension approximations of $\phi$ (Peng et al., 2021; Choromanski et al., 2021; Zhai et al., 2021), the RNN can be implemented as an approximation of the standard attention, which paves the way for the later revival of recurrent models. Such models are nevertheless different from classical RNNs in that the training can be parallel as long as there is no non-linear transform in the transition from $\boldsymbol{z}_t$ to $\boldsymbol{z}_{t+1}$. In addition to simply summing up features as in $\boldsymbol{z}_t$ above, RWKV (Peng et al., 2023) and RetNet (Sun et al., 2023) use exponential time decay of past features. Gating (Yang et al., 2024) or linear transform (Orvieto et al., 2023) have also been introduced. Similarly, a state-space model (SSM) is defined as a transformation of a 1-D input sequence $x(t)$ to output $y(t)$ via an $N$-D hidden state $h(t)$, following

$$h'(t) = \boldsymbol{A}h(t) + \boldsymbol{B}x(t) \tag{2}$$
$$y(t) = \boldsymbol{C}h(t). \tag{3}$$

Hence the discretization of an SSM is akin to an RNN with linear state transition (Gu et al., 2021; 2022). Mamba then renders the transition matrices as $\boldsymbol{A}_t$, $\boldsymbol{B}_t$, and $\boldsymbol{C}_t$ dependent on the input $x(t)$ (Gu & Dao, 2024). Mamba2 (Dao & Gu, 2024) further allows a multi-head formulation in a hardware-efficient way under a generalized framework of various SSMs and linear attention models.

**Linear-complexity models for speech**. With the lower information density or longer form of spoken language, these efficient models have been also introduced to audio and speech processing. There have been attempts to perform ASR using Longformer (Rekesh et al., 2023; Koluguri et al., 2024) and RWKV (An & Zhang, 2023). Also, Mamba has been applied to speech separation (Jiang et al., 2024; Li & Chen, 2024) and enhancement (Li & Chen, 2024; Zhang et al., 2024b). Zhang et al. (2024b) investigated different options of bidirectional Mamba for ASR. Furthermore, there have been attempts to build Mamba-based pretrained speech models (Shams et al., 2024; Yadav & Tan, 2024; Bhati et al., 2024). Such pretraining is nevertheless expensive, especially when taking in to account the need of utilizing the ever-growing set of linear-complexity models.

## 2.2 DISTILLATION AND MODEL CONVERSION

**Knowledge distillation**. Distilling knowledge from teacher models to a student one by matching the output probabilities is a standard approach to compress ML models, including pretrained language models (LMs) (Hinton et al., 2015; Sanh et al., 2019). Intermediate hidden states of the teacher and student can be further aligned by layerwise distillation (Sun et al., 2019; Aguilar et al., 2020). In addition, Brown et al. (2023) have attempted to distill and compress existing pretrained linear-complexity LMs. In these works, parameters from the teacher are also transferred to the student when possible, hence also represent the direct parameter transfer approach.

**Model conversion**. Distinct from model compression, there is also a series of works aimed at converting (a.k.a. uptraining) existing models into new, more efficient ones. Parameter transfer is generally applied, e.g., in the conversion from standard attention to sparse attention (Zaheer et al., 2020), RNN (Kasai et al., 2021), linear attention (Mao, 2022; Mercat et al., 2024), and multi/grouped-query attention (Ainslie et al., 2023). Distillation, or behavior transfer, from the standard transformer as the teacher has been also introduced, starting from early attempts to convert transformers to RNN using teacher outputs or generations (Senellart et al., 2018; Tang et al., 2019). Zhang et al. (2024a) further tried to reproduce the attention matrix of standard attention in linear attention with a loss to align the teacher and student attention weights. Bhati et al. (2024) also used output distillation for pretraining Mamba-based speech models, but without parameter transfer. A concurrent work then combined the distillation on attention weights, outputs, and hidden states (Bick et al., 2024), and

managed to convert large-scale LMs into Mamba with limited computation costs and performance loss. These works are nevertheless focused on conversion during pretraining; by contrast, we focus on conversion during fine-tuning, which is generally cheaper and more accessible. Choi (2024) investigated conversion to RetNet or Mamba during fine-tuning on commonsense reasoning tasks, but using only parameter transfer. Zhang et al. (2024a) also considered the case of conversion after fine-tuning by attention weight alignment. A concurrent work converts Llama3 into Mamba using distillation on outputs and generations during instruction fine-tuning (Wang et al., 2024). Meanwhile, LoSparse jointly fine-tunes and compresses linear layers into LoRA + sparse matrix (Li et al., 2023). Our work is nevertheless distinct in that we investigate different modes of distillation under a general framework of joint fine-tuning and conversion into efficient architectures based on layerwise distillation, which allows a broader range of models and tasks as discussed in Section 1 above.

## 3 METHODS

We aim at building a broadly applicable framework to convert an existing pretrained model into a task-specific linear-complexity one. To achieve this goal, we combine parameter transfer and distillation to best reproduce the performance of the **target teacher model**, a standard transformer model fine-tuned on the target task from the pretrained **source teacher model**. For parameter transfer, we replace only the attention layers in the teacher with the more efficient sequence-mixing modules we intend to use, forming the **student model** as a result; most other layers including embeddings, feed-forward, and layernorm will be preserved. Following this approach, the teacher parameters can be used to initialize the student model as much as possible. This constitutes our baseline **unguided** approach, in which we directly fine-tune the student model with transferred parameters. Instead, by posing loss on the difference of hidden states, we enforce constraints on the deviation of the student's hidden states from the teacher. In this way, we directly guide each layer in the student to reproduce the behavior of the corresponding teacher layer. In addition, we consider different ways or modes of such guidance, introduced below. These modes are also illustrated in Figure 1, and pseudo-codes for the different modes of training algorithms are available in Appendix B.

- **Target Guided**. We can directly transfer the parameters from the fine-tuned teacher target model and distill from it. More specifically, given the model inputs for training on a target classification task, we denote the hidden states from the student and the teacher target as $\boldsymbol{H}^{(s)}$ and $\boldsymbol{H}^{(t)}$, each with $m$ vectors; outputs (class probabilities) of the student and the teacher target as $\boldsymbol{y}^{(s)}$ and $\boldsymbol{y}^{(t)}$; and one-hot labels as $\boldsymbol{y}$. Then the loss terms are written as

$$\mathcal{L}_{\text{CE}}(\boldsymbol{y}^{(s)}, \boldsymbol{y}) = -\sum_i \boldsymbol{y}_i \log(\boldsymbol{y}_i^{(s)}) \tag{4}$$

$$\mathcal{L}_{\text{KD}}(\boldsymbol{y}^{(s)}, \boldsymbol{y}^{(t)}) = \sum_i \left(\frac{\boldsymbol{y}_i^{(t)}}{\beta}\right) \log\left(\frac{\boldsymbol{y}_i^{(t)}/\beta}{\boldsymbol{y}_i^{(s)}/\beta}\right) \tag{5}$$

$$\mathcal{L}_{\text{LD}}(\boldsymbol{H}^{(s)}, \boldsymbol{H}^{(t)}) = \frac{1}{m}\sum_{i=1}^m \left(\boldsymbol{H}_i^{(s)} - \boldsymbol{H}_i^{(t)}\right)^2 \tag{6}$$

$$\mathcal{L} = \alpha_{\text{CE}}\mathcal{L}_{\text{CE}} + \alpha_{\text{KD}}\mathcal{L}_{\text{KD}} + \alpha_{\text{LD}}\mathcal{L}_{\text{LD}} \tag{7}$$

  We set the temperature $\beta = 2$ following Hinton et al. (2015). The final loss $\mathcal{L}$ is a combination of the cross entropy $\mathcal{L}_{\text{CE}}$, the KL divergence $\mathcal{L}_{\text{KD}}$ between teacher and student outputs in standard KD, and an MSE loss $\mathcal{L}_{\text{LD}}$ between hidden states for layerwise distillation. The three terms are weighted by a set of hyperparameters $\alpha$. Instead, the unguided mode uses $\mathcal{L}_{\text{CE}}$ only. More specifically, as the target sequence-mixing module (e.g., RNN) is nevertheless distinct from standard attention, it can be more difficult to reproduce the hidden states right after the sequence-mixing layer. Instead, we extract the hidden states after the processing of the following feed-forward layer as $\boldsymbol{H}$ for layerwise distillation. As for tasks other than classification, we will replace the $\mathcal{L}_{\text{CE}}$ term with the corresponding loss function, e.g. CTC loss for ASR tasks.

- **Trajectory guided**. Models initialized from pretrained parameters perform much better than random initialization when trained on downstream tasks. This can be explained by the rich knowledge internalized into the pretrained model, which may be lost during fine-tuning. It has been found that preserving the knowledge during fine-tuning can be helpful

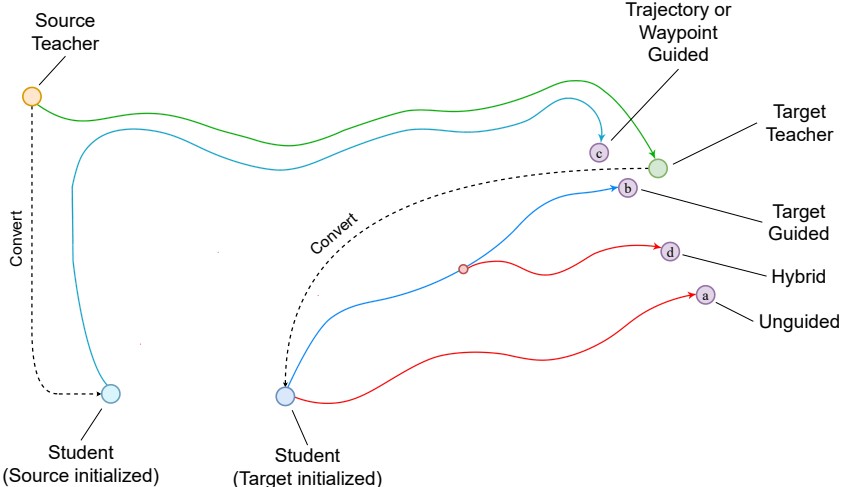

Figure 1: A conceptual illustration of the hidden states shift during training under different modes of distillation. Given the trajectory (green line) of the hidden states during the fine-tuning from the source teacher model (i.e. pretrained transformer) to the target teacher model (i.e. transformer fine-tuned on the target task), we consider: a) Unguided: parameter transfer without any distillation; b) Target Guided: distill from the target teacher; c) Trajectory/Waypoint Guided: gradually distill from a series of models on the trajectory; d) Hybrid: distill from the target teacher until a certain step.

for downstream performance (Li et al., 2018; He & Garner, 2023). Therefore, it can be sub-optimal to only reproduce the hidden states produced by the fine-tuned teacher target model, which has already lost certain knowledge. We are further inspired by Li et al. (2019) who preserve certain hidden states from shifting during fine-tuning for better downstream performance, as well as classical homotopy or continuation methods that solve an optimization problem by tracing a series of surrogate problems that gradually lead to the goal (Allgower & Georg, 1990; Lin et al., 2023). Hence we first initialize the student with source teacher parameters, and then simultaneously fine-tune the teacher and the student. In each step, the distillation loss is not computed against the hidden states of the final teacher target model, but the current teacher model instead. In this way, the student model will reproduce the trajectory of the hidden states during the whole teacher fine-tuning process. As a result, we can better capture the pretraining knowledge present at the early stage of fine-tuning, and adapt to the target task in a way more similar to the teacher. As the converted architecture is different from the original one and may require multiple steps to approximate the current teacher hidden states, we also consider updating the teacher at a slower pace, e.g., every $T_u$ steps.

- **Waypoint guided**. Joint fine-tuning of the teacher and student will essentially double the computation. Instead, we can make use of the checkpoints in teacher fine-tuning. That is to say, we mark the teacher checkpoint of every $T_w$ steps as the guiding *waypoint*. During conversion, we use the nearest waypoint ahead of the current training step as the current teacher. In this way, we approximate the trajectory guided approach in a more coarse-grained way but without extra computation compared to the target guided approach.

- **Hybrid**. The target architecture is after all distinct from the standard transformer, and hidden features in an optimal transformer can be sub-optimal for the target architecture. Therefore, a hybrid scheme between the target guided and unguided modes can be useful. Similar to KD annealing (Jafari et al., 2021), we apply the target guided mode at the initial phase of fine-tuning. We later switch to the unguided mode, which allows the model to train without the distillation constraint. In this way, the initial distillation provides a good initialization for the following optimization.

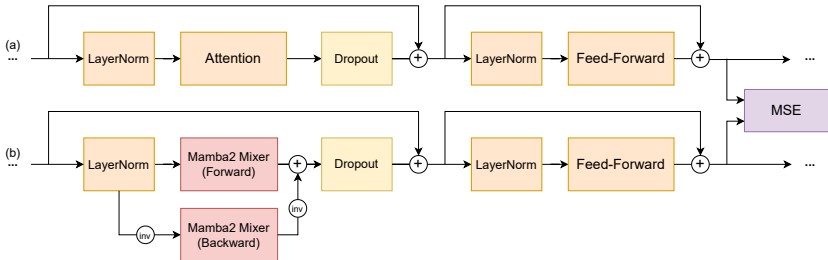

Figure 2: Encoder layers in the student bidirectional Mamba2 model (b), converted from Wav2Vec2 encoder layers in the teacher model (a). In each encoder layer, the attention layer is replaced by two new forward and backward Mamba2 mixers. Inputs and outputs of the backward mixer are timewise inverted. Hidden states after the feed-forward layer are extracted for layerwise distillation.

## 4 EMPIRICAL STUDIES

### 4.1 MODEL CONFIGURATION

We carry out experiments on several scenarios that are representative to demonstrate the advantages of our approach, including

- Converting RoBERTa to Linformer, fine-tuned on language processing tasks, and
- Converting Wav2Vec2 to the latest Mamba2, fine-tuned on speech tasks.

Although not our focus, we also carry out an extra experiment to convert a widely-used open-source LM of various sizes, namely Pythia (Biderman et al., 2023), into Mamba for language modeling by retraining on a small subset of the pretraining corpus.

As mentioned above, we build the target architecture that allows parameter transfer as much as possible. In this way, the architecture may not be identical to the standard Linformer, Mamba, or Mamba2, as specifically described below:

- Language processing by RoBERTa $\rightarrow$ Linformer: We only add the $E$ and $F$ matrices to project the hidden features, which are randomly initialized by $\mathcal{N}(0, 1)$ to preserve the scale of features after projection. All other parts of the model are left unchanged and initialized with pretrained RoBERTa parameters. We follow the reported best Linformer configuration (KV or layer-shared $E$ and $F$, with rank $k = 256$).
- Language modeling by Pythia $\rightarrow$ Mamba: We only replace each attention layer with a Mamba mixer of the same hidden size. The mixer is initialized using the specific Mamba state-space scheme to best preserve the past memory (Gu & Dao, 2024).
- Speech processing by Wav2Vec2 $\rightarrow$ Mamba2: Wav2Vec2 is a bidirectional model, while Mamba2 is unidirectional. Hence we build a bidirectional Mamba2 similar to the one by Zhang et al. (2024b). Each attention layer is replaced by two Mamba2 mixers of the same hidden size, but with expand factor = 1, as shown in Figure 2. The mixer is similarly initialized with the specific Mamba2 state-space scheme (Dao & Gu, 2024).

In this way, all the models have roughly the same number of parameters after conversion. Classification tasks are carried out by adding a linear prediction head after mean pooling. The modeling and training configuration under each scenario is further introduced below, and more details including hyperparameters are provided in Appendix A. Additional details on the computational costs are given in Appendix C.

### 4.2 LANGUAGE PROCESSING

We follow the settings in the Linformer paper to fine-tune the models and report the validation accuracy on a set of widely-used benchmarks: QNLI (Rajpurkar et al., 2016) for natural language

Table 1: Performance of Linformer models converted from RoBERTa, measured by accuracy on benchmarks reported by the original Linformer paper (Wang et al., 2020), in comparison with the reported results of a fully re-pretrained Linformer. The best results are bolded.

|  | QNLI | QQP | SST2 | IMDB | Average |
|---|---|---|---|---|---|
| Pretrained Linformer | 91.2% | 90.8% | 93.1% | 94.1% | 92.3% |
| Std. RoBERTa | 92.4% | 91.8% | 95.3% | 95.7% | 93.8% +1.3 |
| Unguided | 69.4% | 84.3% | 83.6% | 82.6% | 80.0% -12.3 |
| CALD |  |  |  |  |  |
| - Target Guided | 89.0% | 91.8% | 93.3% | 92.3% | 91.6% -0.7 |
| - Src. init. | 88.5% | 91.7% | 93.1% | 92.3% | 91.4% -0.9 |
| - Trajectory Guided | **91.2%** | **91.9%** | **94.0%** | **93.1%** | **92.5%** +0.2 |
| - Waypoint Guided | 89.9% | **91.9%** | 93.7% | 92.8% | 92.1% -0.2 |
| - Hybrid | 86.8% | 90.8% | 91.4% | 90.5% | 89.9% -2.4 |

inference, QQP (Iyer et al., 2017) for text similarity, as well as SST2 (Socher et al., 2013) and IMDB (Maas et al., 2011) for sentiment classification; the former three being part of the GLUE benchmark (Wang et al., 2019). The models are converted from and compared with pretrained RoBERTa-base. In this way, we are able to directly compare with the reported results of the fully re-pretrained Linformer, which is not publicly available. We fine-tune all models for 10 epochs, and a waypoint is taken after each epoch.

The results are given in Table 1. Compared with our fine-tuned standard RoBERTa, the re-pretrained Linformer has a slight drop in performance (1.3% on average). This is typical and can be explained by the reduced expressivity compared to the standard attention. While without extensive pretraining, unguided conversion with only parameter transfer leads to much lower accuracy and severe training instability. Despite much lower computational costs, such performance regression is unacceptable. Fortunately, the regression can be alleviated using CALD. By simply using the target guided approach, the performance drop can be reduced to merely 0.7% on average compared to pretrained Linformer. With the trajectory guided approach, the CALD model can even reach better results (+0.2% on average) but without any pretraining. The approximated waypoint guided approach leads to only slightly lower results, but still better than the target guided one. The hybrid approach leads to lower results compared to target guided, indicating that the guidance is helpful even after early training updates, which matches the rather low unguided results; this is further discussed below in Section 4.4. In addition, we try to use the target guided approach but initialized from the teacher source parameters. Since the distillation should be easier if the student model is initialized with parameters exactly from the teacher model (i.e. the teacher target), these *Src. init.* models show lower results as we expected. All these results indicate that the behavior transfer by CALD is effective and critical for the joint conversion and fine-tuning in this scenario to minimize the performance regression. Particularly, the trajectory or waypoint guided approaches are helpful.

### 4.3 LANGUAGE MODELING

We then present the results of experiments on converting Pythia to Mamba for language modeling. This is nevertheless not our focus as recent pretrained LMs are typically directly evaluated by zero or few-shot performance in downstream tasks without fine-tuning. Therefore, we will essentially need to do the re-pretraining under this scenario. Yet we try to convert the model by training only on a tiny (0.5%, 1%, or 2%) subset of the pretraining corpus, i.e. the deduplicated Pile (Gao et al., 2021; Biderman et al., 2023). Under this scenario, we can only compare the unguided, target guided, and hybrid approaches. Models are trained for 2 epochs. Due to limitations in computational resources, we perform experiments only on Pythia-1B.

We evaluate the models using a set of benchmarks based on those reported in the Pythia paper as shown in Table 2. Compared with the standard Pythia-1B, there remains a performance gap. Nevertheless, we can find that the gap gets limited using 2% data. More importantly, the CALD models consistently get better results than the unguided ones, especially with the hybrid method.

Table 2: Zero-shot performance of downstream evaluation tasks reported by Pythia (Biderman et al., 2023) on Mamba-based language models converted from Pythia-1B, using a small subset of Pile.

| Model | Lambada | PIQA | Winog. | WSC | ArcE | ArcC | SciQ | LogiQA | Avg. |
|---|---|---|---|---|---|---|---|---|---|
| Pythia-1B | 0.562 | 0.707 | 0.535 | 0.670 | 0.570 | 0.244 | 0.839 | 0.221 | 0.544 |
| 0.5% Pile / 1.5B tokens | | | | | | | | | |
| Unguided | 0.394 | 0.671 | 0.493 | 0.542 | 0.502 | 0.211 | 0.778 | 0.246 | 0.479 |
| Tgt. Gd. | 0.410 | 0.686 | 0.504 | 0.608 | 0.538 | 0.210 | 0.775 | 0.230 | 0.495 |
| Hybrid | 0.432 | 0.683 | 0.507 | 0.608 | 0.537 | 0.209 | 0.788 | 0.238 | 0.500 |
| 1% Pile / 3.0B tokens | | | | | | | | | |
| Unguided | 0.453 | 0.673 | 0.514 | 0.571 | 0.525 | 0.217 | 0.794 | 0.217 | 0.495 |
| Tgt. Gd. | 0.449 | 0.689 | 0.518 | 0.626 | 0.535 | 0.220 | 0.798 | 0.218 | 0.507 |
| Hybrid | 0.479 | 0.693 | 0.520 | 0.648 | 0.531 | 0.224 | 0.808 | 0.247 | 0.519 |
| 2% Pile / 6.0B tokens | | | | | | | | | |
| Unguided | 0.475 | 0.687 | 0.535 | 0.612 | 0.543 | 0.222 | 0.802 | 0.237 | 0.514 |
| Tgt. Gd. | 0.459 | 0.697 | 0.527 | 0.656 | 0.547 | 0.218 | 0.817 | 0.235 | 0.520 |
| Hybrid | 0.485 | 0.708 | 0.510 | 0.645 | 0.556 | 0.224 | 0.829 | 0.244 | 0.525 |

Table 3: Performance (%) of Mamba2-converted Wav2Vec2 models: word error rate for ASR on TED-LIUM, as well as accuracy for intent classification on SLURP and speaker ID on VoxCeleb1. The best results are bolded.

| | ASR WER ↓ | IC Acc. ↑ | SID Acc. ↑ |
|---|---|---|---|
| Std. Wav2Vec2 | 6.24 | 91.70 | 96.09 |
| Unguided | 11.29 | 79.68 | 84.24 |
| CALD | | | |
|   - Target Guided | 6.56 | 90.43 | **96.56** |
|   - Waypoint Guided | 6.92 | 90.32 | 96.16 |
|   - Hybrid | **6.41** | **91.23** | 96.41 |

We also observe that the unguided training is rather unstable with frequent NaNs. All the results support the effectiveness of our CALD approach.

## 4.4 SPEECH PROCESSING

We then convert the Wav2Vec2-large models into the latest Mamba2, using a bidirectional architecture specified above. We evaluate them on three common speech tasks with corresponding common benchmarks: ASR with TED-LIUMv3 (Hernandez et al., 2018), intent classification, a standard SLU task, with SLURP (Bastianelli et al., 2020), along with speaker ID with VoxCeleb1 (Nagrani et al., 2020), following the SUPERB configuration (Yang et al., 2021). Specifically, ASR training is performed by CTC loss instead of cross entropy as $L_{CE}$. The converted models are compared with standard fine-tuned Wav2Vec2 models. We take a waypoint every 10,000 steps during fine-tuning. Test word error rate (WER) and accuracy are reported respectively.

Results are given in Table 3, which are consistent with previous findings. There is a significant gap between the performance of standard Wav2Vec2 and the unguided conversion. This can be largely alleviated by CALD using the target guided approach, reaching results close to or even better than standard Wav2Vec2. The hybrid approach further brings slight improvements on ASR and IC.

However, the waypoint guided approach is found to be not helpful or even harmful; we choose not to proceed with the more precise trajectory guided approach. Our assumption is that the trajectory guided approach helps retain the knowledge in the hidden states of the pretrained model. Hence

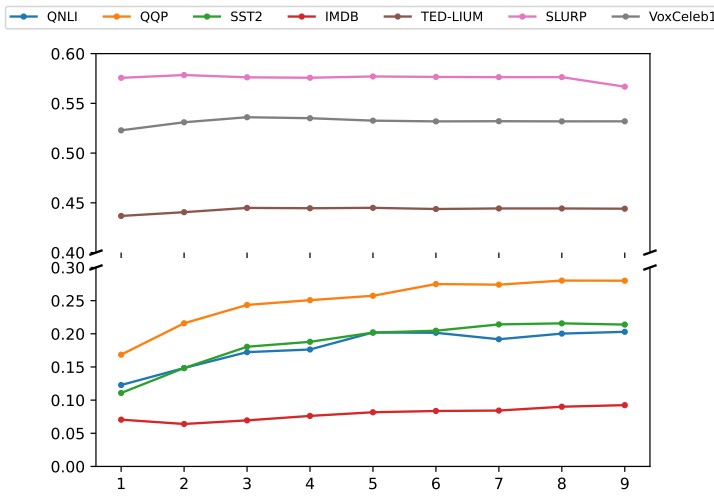

Figure 3: Average cosine distance between the hidden states produced by the initial model and the checkpoints during fine-tuning after each epoch (for NLP tasks) or every 10,000 steps (for speech tasks). Unlike NLP models, features produced by the fine-tuned speech models are far from the initial ones since the early phase of fine-tuning.

We hypothesize that the hidden states of the Wav2Vec2 model undergo significant shift during fine-tuning. As a result, the hidden states of the original pretrained model or those in the early phase of fine-tuning are less useful for the target task. To examine the hypothesis, for each model, we compute the average cosine distance of the hidden states produced by the source teacher and the waypoints, using 100 random data samples. As shown in Figure 3, as for fine-tuning RoBERTa on NLP tasks, the cosine distance is relatively small and increases gradually. While the distance is much larger during the fine-tuning of Wav2Vec2 on speech tasks, even at the early phase of fine-tuning. In fact, the distance has already reached 0.372 after only 2,000 steps of fine-tuning on TED-LIUM. Therefore, our hypothesis is supported that the trajectory guided approach works better in the cases when there is limited hidden state shift during fine-tuning, and when the pretrained model features remain relevant for the target task. While the condition is not satisfied for speech tasks, in which case allowing the model to deviate more from the initial behavior using the hybrid approach will be beneficial.

In addition, we have also attempted to train the models that are unguided and without any parameter transfer. Such a large Mamba2-based speech model trained on the target task from scratch is rather unstable and fails to converge well. This highlights the necessity of pretraining in such large models.

## 4.5 HIDDEN STATE TRAJECTORY

We further visualize the trajectory of hidden states during fine-tuning on the QNLI dataset using different guidance modes by applying t-SNE to concatenated hidden states of 20 random samples in the training set. Hidden states produced by every half-epoch checkpoint are included, hence a total of 20 points during training are plotted for each model. As shown in Figure 4, the actual trajectory matches our expectation conceptualized in Figure 1 that the target guided model approaches the fine-tuned target teacher model (the end of the green trajectory), while the waypoint or trajectory guided models further mimic the trajectory, which is found to be beneficial to the final results. In sharp contrast, the unguided model will completely deviate from the teacher, which leads to training instability and failure to reproduce the original performance.

## 5 CONCLUSION

We investigate the task of converting various existing pretrained transformer models into efficient linear-complexity architectures without the need to redo the whole pretraining. For this goal, we pro-

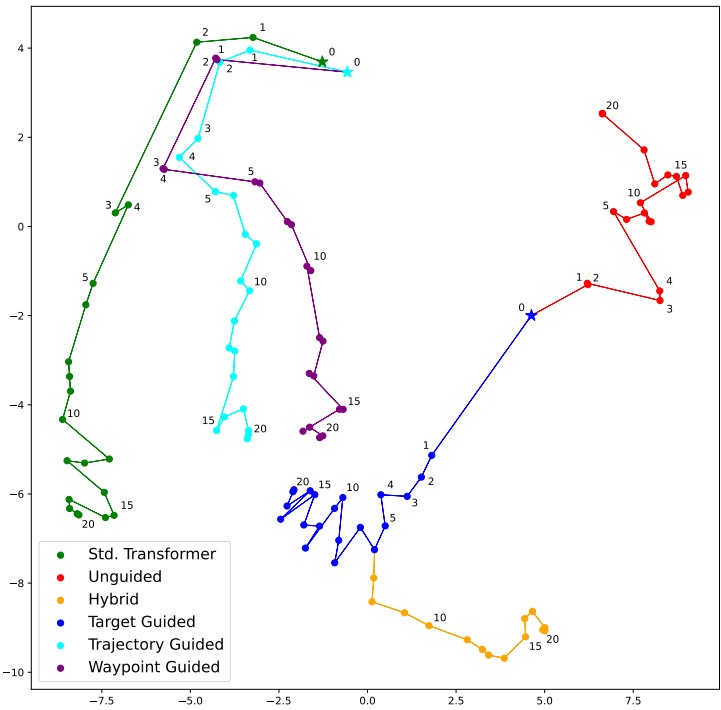

Figure 4: The trajectory of hidden states during training under different modes of guidance, visualized using t-SNE.

pose a novel and broadly compatible Cross Architecture Layerwise Distillation (CALD) approach, and further enhance CALD with a trajectory-based guidance or a hybrid approach. Through our empirical experiments, we confirm that CALD can effectively convert the transformer into an efficient linear-complexity model with performance close to or on par with the standard transformer, much better than directly converted models. We also verify the versatility of CALD by experiments on multiple tasks with various speech and language models. We further identify the tasks where the enhancement approaches are effective in improving the results.

ACKNOWLEDGEMENTS

This work received funding under project SteADI, Swiss National Science Foundation grant 197479.

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

## A    IMPLEMENTATION DETAILS

We build the models as mentioned above, and train the models using the AdamW optimizer. On Mamba/Mamba2 models, we also find it important to apply layer normalization on hidden states prior to computing the layerwise distillation loss, i.e. we take the hidden states after the layernorm operation at the following layer. Otherwise the hidden states will have a rather wide range and bring training instability. We only use output distillation by setting $\alpha_{KD} = 1$ in ASR experiments. On other experiments, we find that the distillation on output probabilities actually leads to suboptimal results, and thus set $\alpha_{KD} = 0$. We always use $\alpha_{CE} = 1$. As for the hybrid mode of distillation, we start with the optimal target guided configuration, but switch to unguided mode when the loss is close to converge, generally at around 30% of the total training steps.

For each target task, we train the models based on its common training recipe for standard transformers, and further search and determine hyperparameters and the model checkpoint to use according to the validation performance. Details for each task are listed below, and more specific details can be accessed from our source code.

- Language processing: We train the model using a learning rate (LR) schedule of linear decay with 6% warmup steps. A total batch size of 32 is used. Other hyperparameters are listed in Table 4.

Table 4: Hyperparameters for language processing tasks.

|  |  | QNLI | QQP | SST2 | IMDB |
|---|---|---|---|---|---|
| Std. RoBERTa | LR | 3e-5 | 3e-5 | 3e-5 | 1e-5 |
| Unguided | LR | 1e-5 | 1e-5 | 3e-5 | 1e-5 |
|  | Share mode | KV | Layer | KV | KV |
| Target Guided | LR | 1e-4 | 1e-4 | 1e-4 | 3e-4 |
|  | $\alpha_{LD}$ | 25 | 20 | 10 | 15 |
|  | Share mode | Layer | Layer | KV | KV |
| Traj. Guided | LR | 3e-4 | 1e-4 | 1e-4 | 3e-4 |
|  | $\alpha_{LD}$ | 30 | 25 | 30 | 40 |
|  | Share mode | Layer | KV | KV | KV |
|  | $T_u$ | 5 | 1 | 3 | 3 |
| Waypoint Guided | LR | 1e-4 | 1e-4 | 1e-4 | 3e-4 |
|  | $\alpha_{LD}$ | 10 | 20 | 5 | 25 |
|  | Share mode | KV | Layer | KV | KV |

- Language modeling: We train the model using a learning rate of 3e-4 and a schedule of cosine decay with 3% warmup steps and a 3e-5 minimum learning rate. We use a total batch size of 128, each with 2048 tokens. We also apply the weight decay of scale 0.1 on Mamba parameters following the Mamba training scheme, along with gradient norm clipped to 1. We set $\alpha_{LD} = 15$ in the experiments. FP16 mixed precision training is adopted.

- Speech processing: We train the model using a learning rate schedule of exponential decay with 5000 warmup steps and 1.5e-6 minimum learning rate. We use a dynamic batch strategy that leads to a batch size of approximately 64 in average. We also apply the weight decay of scale 5e-3, along with gradient norm clipped to 1. For ASR, IC, and SID, we set $\alpha_{LD}$ to 15, 15, and 30, respectively. BF16 mixed precision training is adopted. We find that these hyperparameters work well across different modes of distillation.

## B    DISTILLATION ALGORITHMS

Here we provide the pseudo-code to describe the training algorithms under different modes of distillation to facilitate understanding. Please refer to Section 3 for the notation scheme we use, including the model output terms and loss functions.

## C    COMPUTATIONAL COSTS

The proposed method is presented as a more efficient alternative to re-doing the whole pretraining process on the target architecture, as the model pretraining is notoriously expensive and generally infeasible with academic computation. For example, Wav2Vec2-large pretraining took 5.2 days on 128 V100 GPUs (Baevski et al., 2020), while our target guided CALD experiment to convert Wav2Vec2-large into Mamba2 for ASR takes only 1.6 days on a single RTX3090 with just 24GB memory. Without the computation on the teacher model, the unguided training will be roughly 40% faster, but the accuracy degradation is unacceptable. The hybrid approach is faster owing to more than half of the training steps being under the unguided mode, while reaching better performance than the target guided one. The waypoint guided model doesn't involve any extra computation compared to the target guided one, and hence takes roughly the same time. In NLP experiments, the trajectory guided models enjoy better performance at the cost of more computation. For example, on QNLI with layer-shared Linformer projection, the target or waypoint guided training takes around 3.4 hours under our training configuration with a single RTX3090, while the trajectory guided training takes 3.5 hours ($T_u = 5$) to 4.0 hours ($T_u = 1$). The unguided training takes around 3.0 hours.

---

**Algorithm 1** CALD training under the target guided or hybrid mode.

---

**Require:** `mode`: training mode (hybrid or target guided); $T_h$: number of guided steps in the hybrid mode; $T$: total number of training steps; `dataloader`: loader of training data; $\alpha_{CE}$, $\alpha_{KD}$, $\alpha_{LD}$: hyperparameters for loss term scaling.
1: Build and initialize the teacher model $M_T$ and the student model $M_S$ with the teacher target model.
2: Replace the attention layers in $M_S$.
3: Build the optimizer for $M_S$.
4: **for** $i = 1$ to $T$ **do**
5:     $(x, y) \leftarrow$ `next(dataloader)`
6:     $(y^{(s)}, H^{(s)}) \leftarrow M_S(x)$
7:     **if** `mode = Hybrid` **and** $i > T_h$ **then**
8:        $\mathcal{L} \leftarrow \alpha_{\text{CE}} \cdot \mathcal{L}_{\text{CE}}(y^{(s)}, y)$
9:     **else**
10:        $(y^{(t)}, H^{(t)}) \leftarrow M_T(x)$
11:        $\mathcal{L} \leftarrow \alpha_{\text{CE}} \cdot \mathcal{L}_{\text{CE}}(y^{(s)}, y) + \alpha_{\text{KD}} \cdot \mathcal{L}_{\text{KD}}(y^{(s)}, y^{(t)}) + \alpha_{\text{LD}} \cdot \mathcal{L}_{\text{LD}}(H^{(s)}, H^{(t)})$
12:     **end if**
13:     $\mathcal{L}$.`backward()`
14:     Update $M_S$ parameters using the optimizer.
15: **end for**

---

**Algorithm 2** CALD training under the trajectory guided mode.

---

**Require:** $T_u$: interval between teacher model updates; $T$: total number of training steps; `dataloader`: loader of training data; $\alpha_{CE}$, $\alpha_{KD}$, $\alpha_{LD}$: hyperparameters for loss term scaling.
1: Build and initialize the teacher model $M_T$ and the student model $M_S$ with the teacher source model (the original pretrained model).
2: Replace the attention layers in $M_S$.
3: Build the optimizer for $M_S$ and $M_T$.
4: **for** $i = 1$ to $T$ **do**
5:     $(x, y) \leftarrow$ `next(dataloader)`
6:     $(y^{(t)}, H^{(t)}) \leftarrow M_T(x)$
7:     **if** $i \bmod T_u = 0$ **then**
8:        $\mathcal{L}_T = \mathcal{L}_{\text{CE}}(y^{(t)}, y)$.
9:        $\mathcal{L}_T$.`backward()`
10:        Update $M_T$ parameters using the optimizer.
11:     **end if**
12:     $(y^{(s)}, H^{(s)}) \leftarrow M_S(x)$
13:     $\mathcal{L} \leftarrow \alpha_{\text{CE}} \cdot \mathcal{L}_{\text{CE}}(y^{(s)}, y) + \alpha_{\text{KD}} \cdot \mathcal{L}_{\text{KD}}(y^{(s)}, y^{(t)}) + \alpha_{\text{LD}} \cdot \mathcal{L}_{\text{LD}}(H^{(s)}, H^{(t)})$
14:     $\mathcal{L}$.`backward()`
15:     Update $M_S$ parameters using the optimizer.
16: **end for**

---

---

**Algorithm 3** CALD training under the waypoint guided mode.

---

**Require:** $W$: list of waypoint models; $T_w$: interval between waypoint switching; $T$: total number of training steps; `dataloader`: loader of training data; $\alpha_{CE}$, $\alpha_{KD}$, $\alpha_{LD}$: hyperparameters for loss term scaling.
1: Build and initialize the teacher model $M_T$ with $W_1$, $w_i \leftarrow 1$.
2: Build and initialize the student model $M_S$ from the teacher source model (the original pretrained model).
3: Replace the attention layers in $M_S$.
4: Build the optimizer for $M_S$.
5: **for** $i = 1$ to $T$ **do**
6:     $(x, y) \leftarrow$ `next(dataloader)`
7:     $(y^{(s)}, H^{(s)}) \leftarrow M_S(x)$
8:     $(y^{(t)}, H^{(t)}) \leftarrow M_T(x)$
9:     $\mathcal{L} \leftarrow \alpha_{\text{CE}} \cdot \mathcal{L}_{\text{CE}}(y^{(s)}, y) + \alpha_{\text{KD}} \cdot \mathcal{L}_{\text{KD}}(y^{(s)}, y^{(t)}) + \alpha_{\text{LD}} \cdot \mathcal{L}_{\text{LD}}(H^{(s)}, H^{(t)})$
10:     $\mathcal{L}$.`backward()`
11:     Update $M_S$ parameters using the optimizer.
12:     **if** $i \bmod T_w = 0$ **then**
13:         $w_i \leftarrow w_i + 1$
14:         Re-initialize $M_T$ with $W_{w_i}$.
15:     **end if**
16: **end for**

---

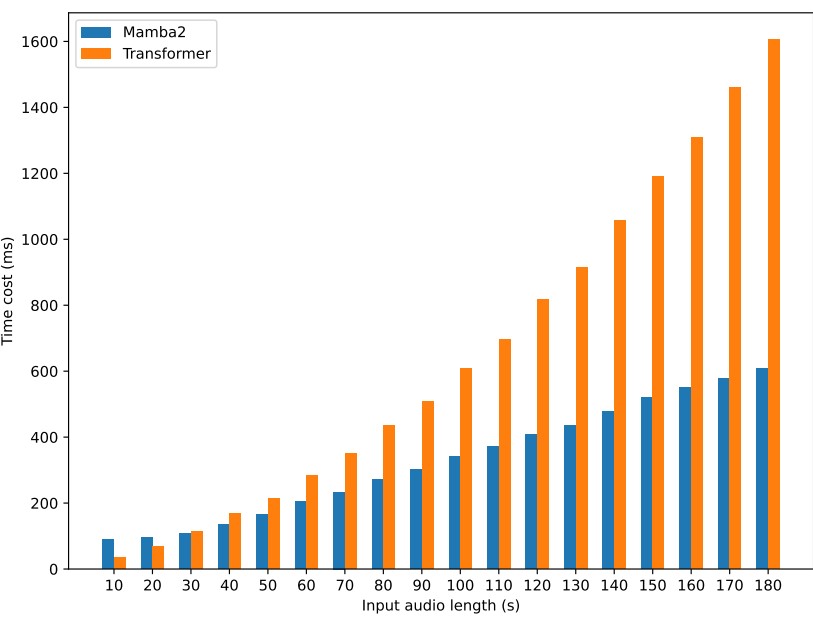

Figure 5: Time costs for Mamba2 and transformer-based models performing ASR on audio samples with different lengths, averaged by 5 runs on a single RTX3090.

To sum up, the CALD approach saves much computation compared to re-pretraining, and the extra cost compared to the unguided training is limited.

We also provide a brief evaluation on the inference speed of transformer-based versus Mamba2-based ASR models we used in Section 4.4, as shown in Figure 5. With asymptotic linear time complexity, the Mamba2 model will be much faster given sufficiently long inputs compared to the quadratic time transformers, although the actual speedup depends on the exact implementation; an investigation on this topic falls out of the scope of this paper.

