# OpenReview forum: "Joint Fine-tuning and Conversion of Pretrained Speech and Language Models towards Linear Complexity"
_ICLR.cc/2025/Conference — ICLR 2025 Poster_

### Official Review · Reviewer_VgsV · 2024-10-23

**Soundness:** 4
**Presentation:** 4
**Contribution:** 4
**Rating:** 8
**Confidence:** 4

**Summary:**

Latests long-context recurrent models such as Mamba2 have demonstrated performance similar to transformer models, even in large scale scenarios, while being more efficient (no quadratic complexity of attention).
However, they need to be retrained from scratch, which could slow down the adoption of their architecture—especially with so many pre-trained transformer models already available.
Moreover, for non-text modalities such as speech, large scale Mamba 2 models are simply not available yet.
Hence authors propose leveraging off-the-shelf pre-trained (transformer) models and converting them into the target (mamba 2) model with linear complexity.
More specifically, they explore the possibility of converting an existing pre-trained transformer into a linear-complexity model for a specific downstream task; approach proposed is called Cross-Architecture Layerwise Distillation (CALD) and combines parameter transfer and distillation.
Multiple scenarios are explored, including converting RoBERTa to Linformer for NLP tasks, Pythia to Mamba for language modeling, and Wav2Vec2 to Mamba2 for speech tasks; converted models show minimal to no performance loss compared to standard transformers while being of linear (instead of quadratic) complexity.

**Strengths:**

-new approach to address model conversion/distillation from transformer (quadratic) architectures to mamba 2 (linear) architectures

-not only applied to written language but also to speech with convincing results (3 types of tasks overall: NPL, LM and Speech)

**Weaknesses:**

-the obtained model is no longer a general purpose model (converted for a specific downstream task)

-some inference speed evaluations would have been a plus (we know that obtained models with linear complexity should be faster though)

**Questions:**

Q: the conversion/distillation is made for a particular downstream task, would it be possible to make it for the pre-trained - general purpose - model ?

Q: can we further speed-up inference through quantization (with added benefits)?

---

> ### Author Response · Authors · 2024-11-25
>
> Thank you so much for your positive feedback and we have revised the manuscript accordingly to address your concerns. We further respond to each concern below:
>
> > the obtained model is no longer a general purpose model
>
> > the conversion/distillation is made for a particular downstream task, would it be possible to make it for the pre-trained - general purpose - model ?
>
> We consider primarily the scenario of obtaining a task-specific model, as our goal is to avoid re-pretraining, which is prohibitive to be carried out in most academic settings due to computational costs and limited access to large-scale datasets. We nevertheless considered the case of converting and re-pretraining a general-purpose large language model (Pythia-1B) using only a small set of the open-sourced dataset, i.e. 0.5% to 2% of Pile. As shown in Section 4.3 and Table 2, the converted models using our CALD approach can reach zero-shot downstream performance better than unguided one and close to the original Pythia-1B model using only 2\% of the pretraining data. This demonstrates that our approach is also applicable to convert the model under a general-purpose scenario, though more exploration under this scenario is left for future work.
>
> > some inference speed evaluations would have been a plus (we know that obtained models with linear complexity should be faster though)
>
> Thank you for the suggestion. Indeed the model with (asymptotic) linear complexity will be faster for sufficiently large N. While the actual speed-up will be highly dependent on the actual implementation. We are focused on a general framework to convert pretrained models into those new models, while efforts to optimize and benchmark the speed of the resultant model (e.g. Mamba) have been carried out by the respective researchers on the specific model. We nevertheless added some inference speed evaluations in Appendix C, which exemplify the speed advantage of Mamba2 for long-form ASR.
>
> > can we further speed-up inference through quantization
>
> We believe that quantization can surely lead to further speed-up and other benefits (e.g. memory saving). However, quantization itself is a different research area and a different set of techniques largely orthogonal to our approach. Hence we do not have any hypothesis regarding the effect of quantization in our approach, and further exploration can be left for future work.

---

> > ### Comment · Reviewer_VgsV · 2024-11-26
> > **acknowledgement of authors' answer**
> >
> > hi, tks for the answers provided
> > they confort me in my evaluation of this good paper
> > i maintain my score to 8

---

### Official Review · Reviewer_QBSN · 2024-11-01

**Soundness:** 3
**Presentation:** 3
**Contribution:** 3
**Rating:** 6
**Confidence:** 3

**Summary:**

In this work, a cross architecture layer wise distillation (CALD) approach is proposed, which includes converting a transformer model to a linear time substitute by replacing attention modules and fine-tuning the new model towards the target task. The fine-tuning is enhanced by knowledge distillation at different levels and stages. The proposed method is examined in both language modeling and speech processing tasks. The results show CALD can effectively narrow the gaps between linear time based substitute and original transformer model on the downstream tasks.

**Strengths:**

- CALD provides an effective and cost-effective approach to build linear complexity based language model by leveraging pre-trained transformer based language model.
- Multiple knowledge distillation methods are proposed and examined.
- The experiments are conducted with different modalities including both speech and text, and different linear complexity transformers. The results show CALD could achieve good results.

**Weaknesses:**

- Experimental descriptions could be improved by providing more information. For example,  in section 4.2, "the models are converted from and compared with retrained RoBERTa-base."  But in the following description about Table 1, "In addition, we try to use the target guided approach but initialized from the teacher source parameters, which shows slightly lower results". Are other models are not initialized from RoBERTa-base?
- More detailed analysis about different distillation methods would be helpful.
- The proposed method could be further improved by including the discussion about linear complexity attention module initialization, which could be very important for the final results

**Questions:**

- How do you choose hyper parameters in equ. 7?
- In Table 1, Src. init results are worse, why?
- Hybrid approach is better in Table 2 and 3, but it is worse in Table 1. Any explanation?  The trajectory or waypoint guided approaches are worse than the target Guided method in Table 3 but better in Table 1. Any explanation? Has the model learnt enough from the waypoint model?

---

> ### Author Response · Authors · 2024-11-25
>
> Thank you so much for your feedback and we have revised the manuscript accordingly to address your concerns. We further respond to each concern below:
>
> > Are other models are not initialized from RoBERTa-base?
>
> > In Table 1, Src. init results are worse, why?
>
> As mentioned in Sec. 3 and illustrated in Figure 1, the trajectory/waypoint guided models are initialized from the original pretrained (not fine-tuned) RoBERTa-base, while the target guided models are initialized from (and distilled towards) the fine-tuned RoBERTa-base. Empirical results suggest that transferring the fine-tuned transformer parameters leads to better performance compared to the "Src. Init." case when we directly transfer the original non-fine-tuned parameters. This is expected as the teacher model is the fine-tuned one, and it is surely better to initialize the student model using the teacher parameters. We have revised the manuscript to clarify that.
>
> > More detailed analysis about different distillation methods would be helpful.
>
> The distillation methods we used are elaborated in Section 3. In response to another review we added the pseudo-code for our algorithms in Appendix B; we hope it also serves an answer to the present comment.
>
> > including the discussion about linear complexity attention module initialization
>
> We follow the standard way to initialize the Linformer and Mamba for optimal results. As for Linformer, the E and F projection matrices are initialized with N(0,1) to preserve the scale after the projection. As for Mamba, we follow their specialized HiPPO-based initialization to best preserve the past memory. The discussion is added in Section 4.1.
>
> > How do you choose hyper parameters in equ. 7?
>
> We keep $\alpha_{CE} =1$, while $\alpha_{LD}$ is decided by a grid search. As mentioned in Appendix A, we find that the output distillation term is not helpful in tasks other than ASR in our preliminary experiments, which is expected as the LD term is already applied to the previous layers. Hence we set $\alpha_{KD}=1$ in ASR, and 0 in other tasks.
>
> > Hybrid approach is better in Table 2 and 3, but it is worse in Table 1. Any explanation? The trajectory or waypoint guided approaches are worse than the target Guided method in Table 3 but better in Table 1. Any explanation?
>
> Thank you for pointing this out and actually we have observed and discussed about this phenomenon. Please refer to the explorations in Section 4.4 (L430~465) and Figure 3, which are meant to address this. Extra clarifications are added to Section 4.2 to avoid such confusion.

---

> > ### Comment · Reviewer_QBSN · 2024-11-26
> > **Thanks for your replies**
> >
> > I will keep the original score.

---

### Official Review · Reviewer_L6QU · 2024-11-03

**Soundness:** 3
**Presentation:** 3
**Contribution:** 3
**Rating:** 8
**Confidence:** 3

**Summary:**

This paper introduces a method called Cross-Architecture Layerwise Distillation (CALD). The goal of CALD is to convert existing pre-trained transformer models into linear-complexity models and also fine-tune tje,, making them more computationally efficient without the need for extensive re-pretraining. This approach enables the efficient adaptation of models to different architectures, such as transforming RoBERTa into Linformer for NLP tasks and Wav2Vec2 into Mamba2 for speech processing tasks. CALD combines parameter transfer, where attention layers are replaced with efficient sequence-mixing modules, with knowledge distillation, where the student model learns from the teacher model's behavior. Four distillation modes are presented: Target Guided, Trajectory Guided, Waypoint Guided, and Hybrid.
The paper highlights that CALD effectively minimizes performance loss compared to directly converted models, demonstrating its efficacy across various language and speech processing tasks.

**Strengths:**

1.) The introduction of the Cross-Architecture Layerwise Distillation (CALD) is noteworthy, as it aims to convert pre-trained transformer models into efficient linear-complexity architectures. This bridges a critical gap by enabling the reuse of existing pre-trained models without the need for resource-intensive pretraining, especially for non-text domains like speech.

2.) The empirical studies are well-structured and cover various conversion tasks from RoBERTa to Linformer for NLP tasks, Wav2Vec2 to Mamba2 for speech tasks, and Pythia to Mamba for language modeling. The detailed experiments provide a strong basis for assessing the effectiveness of CALD, showing that guided approaches outperform unguided ones significantly.

3.) The paper does a thorough job of comparing the proposed method with existing approaches, including both guided and unguided methods. The results highlight that CALD, especially with trajectory-guided or waypoint-guided distillation, can effectively maintain or improve performance close to standard transformer models.

4.) The inclusion of diverse benchmarks (e.g., QNLI, QQP, TED-LIUM, SLURP, VoxCeleb1) and the report of performance improvements in areas like word error rate (WER) and accuracy provide strong evidence for the robustness of the proposed methodology.

5.) The paper provides a solid theoretical explanation for why trajectory-guided distillation can retain pre-training knowledge and ensure better downstream task performance.

**Weaknesses:**

Limited Discussion on Limitations: While the results are strong, the paper lacks an in-depth discussion on potential limitations or failure cases of the CALD approach, especially in more complex or resource-constrained real-world scenarios.

Computational Cost Analysis: Although the proposed method is presented as a more efficient alternative, there is insufficient analysis of the actual computational cost savings compared to traditional pre-training or conversion processes. A more detailed breakdown of memory and time requirements would enhance the practical relevance.

**Questions:**

Q1 - How does CALD compare with other recent state-space models in terms of accuracy, inference time, and training stability, particularly for large-scale speech tasks?

Q2 - Have the authors tested CALD in real-world deployment scenarios for speech and language tasks? If so, how does its performance hold up compared to controlled experiments?

---

> ### Author Response · Authors · 2024-11-25
>
> Thank you so much for your positive feedback and we have revised the manuscript accordingly to address your concerns. We further respond to each concern below:
>
> > potential limitations or failure cases of the CALD approach, especially in more complex or resource-constrained real-world scenarios.
>
> Thank you for the suggestion. We would like to clarify that the CALD approach is particularly designed for real-world resource-constrained scenarios when we do not have the data and computational resources to re-do the pretraining for each new architecture. CALD will surely fail in more extreme cases when little data and computation are available, while in such cases re-pretraining will be even more infeasible.
>
> > computational cost analysis
>
> The proposed method is presented as a more efficient alternative to re-doing the whole pretraining process on the target architecture, as the model pretraining is notoriously expensive and generally infeasible with academic computation. For example, Wav2Vec2-large pretraining took 5.2 days on 128 V100 GPUs, while our distillation (target-guided) experiments to convert Wav2Vec2-large into Mamba2 for ASR takes only 1.6 days on a single RTX3090 with merely 24GB memory. Without the computational costs on the teacher model, the unguided models will be roughly 30\% faster, but the accuracy degradation is considerable; the hybrid approach enjoys the merits of both approaches. Regarding the computational costs compared to standard transformer models, we further performed some exemplary inference speed evaluations. Nevertheless, the speed optimization of the specific target architecture highly depends on the implementation and falls out of our scope of a generally applicable conversion framework. Relevant discussions are added to Appendix C.
>
> > compare with other recent state-space models...for large-scale speech tasks
>
> Minus pretraining, the model architecture used in our experiments is roughly the same as other recent works using bidirectional Mamba on speech (e.g. [arxiv:2405.12609](https://arxiv.org/abs/2405.12609) ), hence the speed and performance will be similar. However, large-scale pretraining is critical for training larger models, e.g. as large as Wav2Vec2-large used in our experiments. We find that if we reinitialize all the parameters (which renders our model similar to other speech state-space models trained from scratch, but much larger), it will be difficult for the model to converge and the performance will be even lower than the unguided models. This provides a lens to compare with other recent speech state-space models trained from scratch and emphasizes the importance of pretraining. We have added this explanation to Section 4.4 in our manuscript.
>
> > real-world deployment scenarios
>
> We tested CALD on standard benchmarks that represent multiple real-world scenarios, including ASR, intent classification, and speaker ID, where strong performance is demonstrated. However, actual deployment demands more resources that are rather infeasible under academic settings, thus we choose to leave it for future work.

---

### Official Review · Reviewer_hriZ · 2024-11-04

**Soundness:** 2
**Presentation:** 3
**Contribution:** 2
**Rating:** 5
**Confidence:** 4

**Summary:**

This paper focuses on the issue of the quadratic computational complexity of standard self-attention in transformer models and proposes a way to convert pre-trained transformer models with quadratic attention to more efficient linear attention, while trying to minimize performance degradation. They call this method Cross-Architecture Layerwise Distillation (CALD), and the core of this approach is the combination of layer-wise distillation with parameter transfer, making the fine-tuning process more efficient, and a few specially designed fine-tuning strategies.

The authors propose a few strategies to compare with a simple "unguided" approach. They report on language and speech tasks, and show that their proposed methods consistently outperform the unguided approach, and in some cases, some methods can even outperform linear-attention models that are repretrained.

**Strengths:**

The paper is mostly well-written and the problem it tries to solve is an important one. It does a good job reference related works. Novelty-wise, although the individual components of the proposed method aren't new, it has a fair amount of novelty for combining them and designing different fine-tuning strategies for training. The figure showing the actual trajectory of hidden states is interesting and informative to show how parameters change in those different methods.

**Weaknesses:**

I'm not fully convinced of the claimed merits of the method. While it seems to be convincing that those proposed methods are better than unguided approach, there are unanswered questions regarding those approaches, because it seems to have different behaviors in different tasks. E.g. in experiments reported in Table 1, unguided's performance significantly lags behind other methods, while in Table 2, the relative difference is much smaller. This might indicate that the unguided hyper-params might not be well-tuned for certain tasks. Also, in Table 1, the "hybrid" approach gives the worst performance among all CALD variants, but it seems to give the best performance in many of the cases in Table 2 and 3. This to me suggests that there might be some other fundamental factors that caused those different behaviors shown in those Tables.

There are a couple of misreferences in the writing, where the authors wanted to reference Figure 1 but said Figure 3 instead. There's a reference of Figure 4.1 and Figure 4.4 which I believe are typos that point to other Figures in the paper. In section 2.2, there's an incomplete sentence "an example of the direct parameter transfer approach."

It took me a while to fully understand Figure 1, and I think the source of the confusion comes from that the diagram involves both shifts in hidden states and also relapse of time, like the "trajectory/waypoint guided" blue arrow actually "follows" the green arrow as time goes by. I understand the authors might want to show this diagram here to be consistent with Figure 4 shown later in the analysis, but my feeling is this diagram isn't the clearest way to demonstrate the difference between the approaches. I would feel that providing simple pseudo-code instead of this diagram + lots of text in method bullet points might be a better way to accurately represent the methods.

For ASR tasks, the authors use CTC loss instead of CE. How that does change Equation 4? E.g. do you still compute the CE per-frame or some other computation? Also, since CTC adopts a blank token in its output and we have research works that show a CTC model would predict blanks for most frames, I feel there should be some interactions with this aspect instead of treating blanks and non-blanks equally.

**Questions:**

Please see the above "weakness" section.

---

> ### Author Response · Authors · 2024-11-25
>
> Thank you so much for your feedback and we have revised the manuscript accordingly to address your concerns. We further respond to each concern below:
>
> > in experiments reported in Table 1, unguided's performance significantly lags behind other methods, while in Table 2, the relative difference is much smaller
>
> On the one hand, the proposed distillation methods indeed have different effects in different tasks. The Table 2 corresponds to the scenario of zero-shot inference on large language models. In this case, to produce meaningful (not random) inference, the model capacity and training dataset need to be sufficiently large. As we often observe in such large-scale training, the help brought by the enhanced training technique will be reduced compared to the scenario of fine-tuning a smaller model using limited data. This leads to the smaller relative difference in Table 2 compared to Table 1 and Table 3.
>
> On the other hand, we did find issues in hyperparameter tuning in certain tasks and thank you for pointing out that. We performed  grid search of hyperparameters in all the experiments for fair comparison, and as for the QNLI and QQP tasks the unguided model hyperparameters we identified failed to converge very well, leading to rather low accuracy. However, we performed more complete hyperparameter search on all the tasks after the initial submission and found some configurations with better results on QNLI, QQP, and SST2, given in the revised Table 1. The CALD models still outperform the unguided models by more than 10% in average accuracy. Therefore, the conclusions we drawn from the experiments are not affected.
>
> > the "hybrid" approach gives the worst performance among all CALD variants, but it seems to give the best performance in many of the cases in Table 2 and 3
>
> Thank you for pointing this out and actually we have observed and discussed about this phenomenon. Please refer to the explorations in Section 4.4 (L430~465) and Figure 3, which are meant to address this. Extra clarifications are added to Section 4.2 to avoid such confusion.
>
> > There are a couple of misreferences in the writing
>
> Thank you for pointing this out. Those issues are fixed in the revised manuscript.
>
> > I would feel that providing simple pseudo-code instead of this diagram + lots of text in method bullet points might be a better way to accurately represent the methods.
>
> Thank you for the suggestion. Pseudo-codes for the algorithms are added in Appendix B to facilitate understanding.
>
> > For ASR tasks, the authors use CTC loss instead of CE. How that does change Equation 4?
>
> This seems to be a result of our unclear presentation. We will replace the CE loss term with CTC (or any other task specific loss) on tasks other than classification. We added extra explanation in the revised Section 3 to clarify that.
>
> > I feel there should be some interactions with this aspect instead of treating blanks and non-blanks equally
>
> We agree that in such an imbalanced classification scenario, treating the more frequent blank labels in a specialized way can be sensible to facilitate CTC-based ASR in general. However, this technique will be orthogonal to the model conversion methods investigated in this paper. Therefore the investigation of it may fall out of the scope of this paper, and at present we do not have a hypothesis on using this technique properly.

---

### Meta-Review · Area_Chair_Lh86 · 2024-12-20

**Metareview:**

This paper proposes a approach based on distillation for efficient fine-tuning, with an advantage of linear complexity.

All reviewers praise the idea interesting, the paper easy to follow, and the experiments solid across multiple modalities and benchmarks.

Despite the recommendation of acceptance, the paper is not without weaknesses. The paper is conceptually not a big departure from prior work; hence the recommendation of a poster. There are valuable suggestions provided by the reviewers, and I encourage the authors to incorporate the feedback to improve the paper.

**Additional Comments On Reviewer Discussion:**

The authors answered the questions raised by the reviewers. There weren't much discussion beyond that.

---

### Decision · Program_Chairs · 2025-01-22

Accept (Poster)